# Randomized Gradient Subspaces for Efficient Large Language Model Training

## Abstract

Training large language models (LLMs) in single or limited node settings is often bottlenecked by extreme memory demands, with optimizer states dominating the footprint. Recent works mitigates this cost by projecting gradients into low-dimensional subspaces using sophisticated update strategies. In this paper, we analyze the dynamics of gradient space and its underlying subspaces. We find that while a small subspace captures most gradient energy, a significant portion still resides in the residual bulk; moreover, the influence of the core subspace diminishes over time and in deeper layers. We also observe that the gradient space exhibits near-flat curvature, calling for algorithms that explicitly account for this geometry. Motivated by these insights, we introduce a suite of randomized algorithms, GrassWalk and GrassJump , which exploit subspace and achieve state-of-the-art memory savings while improving performance on LLaMA-1B and LLaMA-7B pretraining.

## 1 Introduction

Training large language models (LLMs) demands extreme computational and memory resources, with a significant portion related to optimizer states. A promising line of work reduces this cost by exploiting the observation that gradients evolve in a low-dimensional subspace (Gur-Ari et al., 2018; Schneider et al., 2024). Recent methods project gradients into such subspaces, reducing the size of the optimizer state and memory while still updating all parameters (Zhao et al., 2024a; Robert et al., 2025; Anonymous, 2025; Chen et al., 2025b; Zhu et al., 2025). Structured approaches such as GaLore (Zhao et al., 2024a), Fira (Chen et al., 2025b), LDAdam (Robert et al., 2025) and SubTrack++ (Anonymous, 2025) estimate the subspace using SVDs, PowerSGD or Grassmannian optimization. In contrast, other alternatives such as APOLLO (Zhu et al., 2025) and FRUGAL (Zmushko et al., 2025) rely on random projections to avoid computation cost, while GoLore (He et al., 2025) utilizes randomness in the latter iterations to improve convergence.

While recent works have alleviated some limitations of low-rank gradient methods, through strategies that recover lost information (Chen et al., 2025b; Zmushko et al., 2025; Zhu et al., 2025; Anonymous, 2025) or techniques that adjust optimizer states when the coordinates change (Xiao et al., 2025; Robert et al., 2025; Zmushko et al., 2025; Anonymous, 2025), they do not fully explain why certain randomized strategies succeed or fail. By analyzing the subspace structure of gradients, we clarify this behavior and, importantly, guide the design of more effective randomized algorithms. This perspective reframes the open question: what role can randomized algorithms play in the efficient training of LLMs, once we account for their underlying gradient subspaces?

Analyzing gradient dynamics, we observe that while a low-rank subspace captures most of the gradient energy early on, its share declines over time and is markedly smaller in deeper layers; though it remains non-negligible. This suggests that as training progresses, particularly in later layers, an increasing fraction of learning occurs outside the core subspace. Moreover, we observe that this core gradient subspace often evolves in a nearly flat curvature, underscoring the need to account for this structure when exploiting gradient information. This perspective not only clarifies why certain random-based methods succeed, but also highlights opportunities to design more principled and efficient training algorithms. Based on these insights, and through controlled interventions, we show that our proposed methods, GrassWalk and GrassJump, can achieve state-of-the-art results when de-

signed with awareness of gradient dynamics and the underlying optimization landscape. Ultimately, the key is to exploit all available information and not mislead the optimizer.

Specifically, in this paper, we investigate these questions by analyzing (i) different subspace update methods, (ii) the benefits of projection-aware optimizers (Robert et al., 2025; Anonymous, 2025; Zmushko et al., 2025; Xiao et al., 2025), and strategies for recovering information lost during low-rank projection (Chen et al., 2025b; Zhu et al., 2025; Anonymous, 2025; Zmushko et al., 2025), all through the lens of **gradient subspace dynamics**. Based on these experiments, we introduce Grass-Walk and GrassJump , which apply random walks and random jumps on the Grassmannian manifold to update the underlying subspace, while simultaneously adapting the optimizer to subspace changes and restoring information lost in projection, before each weight update. Our proposed methods consistently outperform strong baselines, matching GaLore-like memory efficiency while delivering superior performance and faster convergence on LLaMA-1B and LLaMA-7B pretraining.

**Our key contributions:**

- We provide a comprehensive analysis of gradient subspaces during LLaMA pretraining, revealing (i) the diminishing dominance of low-rank subspaces over time, especially in deeper layers, and (ii) their evolution in a nearly flat curvature.
- By connecting randomized projections to gradient subspace dynamics, we establish principles that clarify when and why randomization can be effective, offering the first systematic explanation of these methods' strengths and limitations.
- We introduce two methods, GrassWalk and GrassJump , that perform random walks and jumps on the Grassmannian manifold, achieving superior convergence and accuracy.

## 2 LOW-RANK GRADIENT METHODS

Training large-scale models such as LLMs places extreme demands on both computation and memory, with optimizer states often consuming much more memory than the parameters. Gradient low-rank methods address this bottleneck by exploiting the observation that gradients during training often evolve in a low-dimensional subspace of the full parameter space (Gur-Ari et al., 2018; Schneider et al., 2024; Zhao et al., 2024a). They reduce memory usage while supporting full-parameter updates by projecting the gradients $G_t \in \mathbb{R}^{m \times n}$ into a subspace of dimension $r \ll m, n$ (Zhao et al., 2024a; Robert et al., 2025; Zhu et al., 2025; Chen et al., 2025b; Anonymous, 2025), as shown in equation 1. where $S_t \in \mathbb{R}^{m \times r}$ is an orthonormal basis that spans the subspace. In this paper, we assume $m \leq n$ without loss of generality.

$$\widetilde{G}_t = S_t^\top G_t \tag{1}$$

Optimizers such as Adam are then applied in this reduced space, and the results are mapped back to the full parameter space for weight updates. This design yields a significant reduction in the optimizer state size from $O(2mn)$ to $O(mr + 2nr)$ (Zhao et al., 2024a).

Assuming the gradients lie in an underlying low-rank subspace with a known rank $r$, a natural approach is to compute the SVD of the gradient matrix and construct its rank-$r$ approximation (Zhao et al., 2024a), since SVD inherently provides such an estimation, as shown in equation 2.

$$G_t = U_t S_t V_t^\top \approx \sum_{i=1}^{r} s_t^i u_t^i v_t^{i^\top}, \quad S_t = [u_t^1, \ldots, u_t^r] \in \mathbb{R}^{m \times r}. \tag{2}$$

Projecting gradients into a rank-$r$ subspace offers an effective way to reduce an optimizer's memory footprint while still enabling full-parameter tuning. Unlike low-rank weight adaptation methods such as LoRA (Hu et al., 2021), this approach is applicable to both pre-training and fine-tuning. However, it also introduces several challenges.

**Gradient subspaces are inherently unstable.** Although prior works suggest the existence of a low-rank core subspace in the gradient space, they also show that this subspace is not always stable and that its variations must be captured (Zhao et al., 2024a). Several strategies have been developed for updating the low-rank subspace; methods such as GaLore (Zhao et al., 2024a) and FiRA (Chen et al., 2025b) periodically compute the SVD of the gradient matrix to identify the dominant directions, however, SVD is computationally heavy and sensitive to noise (Anonymous, 2025; Zhu et al., 2025;

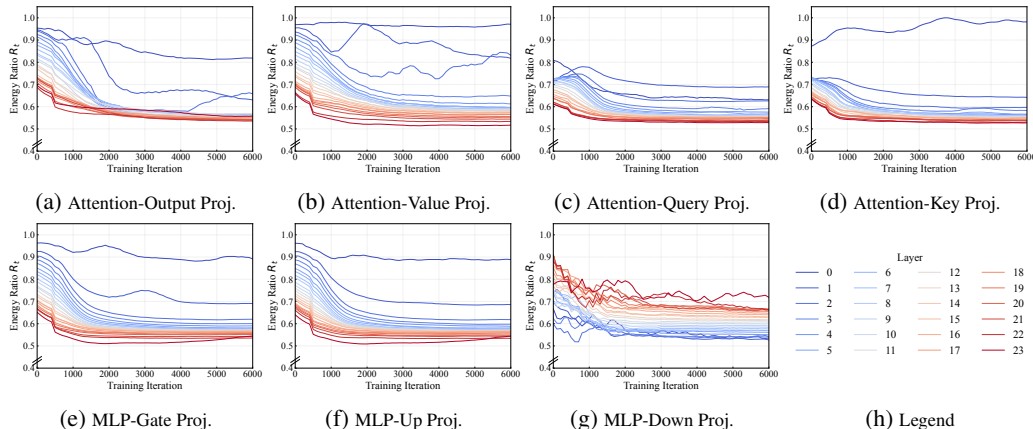

(a) Attention-Output Proj.   (b) Attention-Value Proj.   (c) Attention-Query Proj.   (d) Attention-Key Proj.

(e) MLP-Gate Proj.   (f) MLP-Up Proj.   (g) MLP-Down Proj.   (h) Legend

Figure 1: Each decoder layer stack includes seven layer types in the Llama-1B model. The plots show the fraction of gradient-matrix energy explained by a rank 512 approximation. Despite a high lower bound, this fraction declines over training, and deeper layers generally exhibit smaller fractions.

Robert et al., 2025), particularly in later training stages when the gradients are small, leading to instability and slower convergence (He et al., 2025). To mitigate these drawbacks, methods such as APOLLO (Zhu et al., 2025) and FRUGAL (Zmushko et al., 2025) leverage random projections to reduce computational cost. On the other hand, LDAdam (Robert et al., 2025), Online Subspace Descent (Liang et al., 2024), and SubTrack++ (Anonymous, 2025), employ iterative approximation techniques or subspace tracking to estimate dominant gradient subspaces.

**Momentum states misalign under changing subspaces.** Common optimizers such as AdamW assume a fixed coordinate system and update their internal states accordingly. As a result, momentum states may become misaligned whenever the subspace is updated. To mitigate this issue, COAP (Xiao et al., 2025) proposes aligning subspace updates with the first momentum direction, thereby leveraging information already encoded in the optimizer states. FRUGAL (Zmushko et al., 2025), in contrast, addresses this by either projecting the old states into the new subspace or resetting the momenta altogether upon subspace adjustment. However, simple projections cannot be directly applied to Adam states, as it is not limited to linear operations. To handle this, methods such as LDAdam (Robert et al., 2025) explicitly account for the nonlinearity. In particular, LDAdam reformulates Adam's state as a statistical estimator of the first and second momentum of the gradient matrix and introduces an alternative formulation to address this challenge.

**Low-rank Projections sacrifice gradient information.** A low-rank projection discards gradient components orthogonal to the chosen subspace, thereby losing potentially useful training signals. LDAdam (Robert et al., 2025) addresses this by introducing an error-feedback mechanism that feeds the discarded gradient back into the model in the next iteration. FRUGAL (Zmushko et al., 2025) instead adapts state-free optimizers to handle the discarded portion of the gradients. Building on a different insight, Fira (Chen et al., 2025b) observed that the scaling factor remains consistent across both the dominant and residual subspaces. Leveraging this property, it employs the scaling factor computed by a stateful optimizer (AdamW) to rescale the discarded components before applying weight updates. SubTrack++ (Anonymous, 2025) adopts a similar approach, enabling stateful updates without incurring additional optimizer memory costs.

## 3 IS THERE A CORE SUBSPACE?

Prior work Gur-Ari et al. (2018); Yaras et al. (2023) shows that during gradient descent, gradients evolve within a low-dimensional subspace that carries most of the signal. However, recent work Song et al. (2025) argues that learning is not confined to this core subspace and that constraining updates to it can hinder progress. To study gradient behavior in LLM training, we adopt the SubTrack++ (Anonymous, 2025) setting to leverage a geometrically principled notion of a "core" subspace and its evolution. We then quantify the fraction of gradient energy preserved by the low-

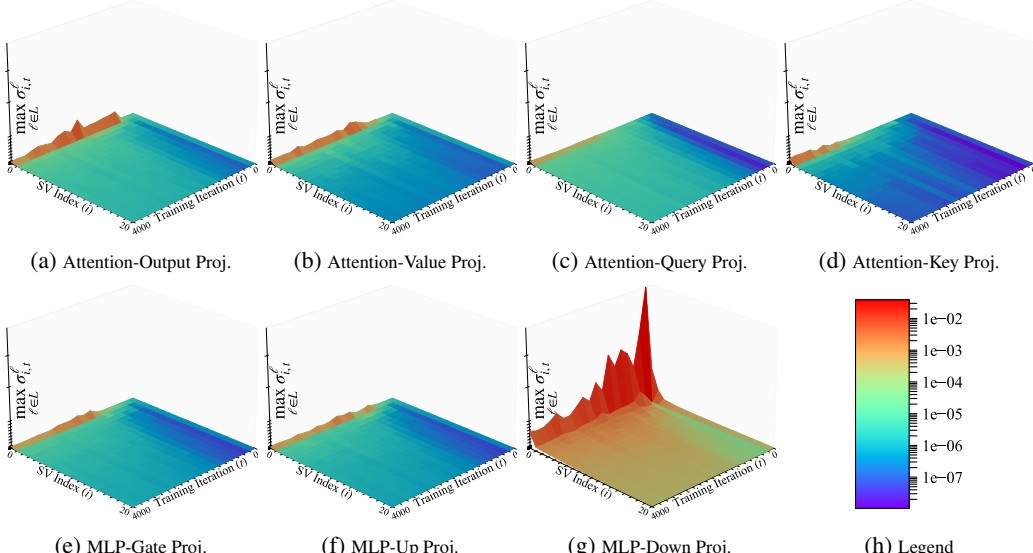

(a) Attention-Output Proj.  (b) Attention-Value Proj.  (c) Attention-Query Proj.  (d) Attention-Key Proj.

(e) MLP-Gate Proj.  (f) MLP-Up Proj.  (g) MLP-Down Proj.  (h) Legend

Figure 2: Evolution of the top 20 singular values of the subspace estimation error derivative across different projection layers in the LLaMA-1B architecture. Each plot shows the maximum $i$-th singular value within a given layer type (aggregated across all 24 decoder layers) as training progresses. While MLP down-projection layers exhibit the largest singular values (g), their magnitude remains small and decays rapidly. Other projection layers (a-d) show values close to zero throughout training. The overall distribution of singular values becomes more uniform as training advances, suggesting that the gradient subspace evolves in an almost flat curvature.

rank approximation by computing the ratio $R_t$ in equation 3, where the Frobenius norms of the low-rank gradient equation 1 and the full-rank gradient are used to calculate this ratio.

$$R_t \;=\; \frac{\|\widetilde{G}_t\|_F}{\|G_t\|_F} \tag{3}$$

We computed this ratio across all layers during training using a LLaMA-based architecture with 1B parameters (Touvron et al., 2023) and with subspace rank $r = 512$. The ratio for training with subspace ranks 256 and 1024 are also reported in Appendix A. The model consists of 24 decoder layers, each containing an attention block followed by an MLP block. The low-rank structure applies to the linear projections, of which there are seven per decoder layer. In the attention block, the key, query, and value projections generate the essential attention representations, while the output projection aggregates the resulting information from all heads. In the MLP block, the up projection expands the hidden representation, the gate projection modulates which features are suppressed or retained, and the down projection compresses the result back to the model dimension. To provide more detailed insights, we clustered the results according to these projection types across all 24 decoder layers.

As shown in Figure 1, across all layer types more than 50% of the gradient energy lies in the core subspace. However, in most layers this fraction decreases as training progresses. The decline is most pronounced in deeper layers, where the fraction is consistently lower, suggesting that during pre-training the subspaces of later layers vary more rapidly and play a less "core" role. This aligns with the common view of LLM pre-training: early layers quickly learn broad, shared features, while continued training shifts capacity toward rarer, more specialized features captured by later layers.

Despite the observed gradient energy, it is essential to understand the space in which we aim to find the optimal projection. To this end, at each subspace update step we compute the derivative of the subspace estimation error with respect to the underlying subspace. This derivative specifies the update direction on the manifold to reduce the estimation error. For every layer in the LLaMA-1B architecture, we then extract the top 20 singular values of this matrix. Figure 2 shows how the distribution of these singular values evolves during training across different layer types.

At each iteration, we report the maximum $i$-th singular value within each layer cluster (e.g., Attention Query or MLP up-projection), aggregated across all 24 decoder layers, thus providing an upper-bound distribution. Notably, even in the MLP down-projection (Figure 2-g), where the largest singular values appear, the range remains small and decays rapidly. For other layer types, the largest singular values are already close to zero. In addition, as training progresses, the singular value distribution becomes increasingly uniform, despite the overall magnitudes being extremely small. Taken together with the results in Figures 1 and 2, these observations suggest that the gradient subspace evolves within an almost flat curvature, while there is a considerable portion of energy carried by the bulk component of the gradient. As in other optimization landscapes, such flatness implies that random steps can be advantageous, preventing the model from being trapped in sharp local optima.

## 4 GRASSWALK AND GRASSJUMP

The Grassmannian manifold, denoted $Gr(r, n)$, is the space of all $r$-dimensional subspaces of an $n$-dimensional space (Bendokat et al., 2024). By definition, when we are using low-rank gradient methods, we aim to find a point on $Gr(r, n)$ onto which the gradients are projected. Building on our analysis of gradient subspace dynamics, we introduce two randomized methods, GrassWalk and GrassJump. These methods generate diverse subspaces through random exploration, preventing the optimizer from being trapped in flat regions of the manifold. At the same time, they leverage all available gradient information by retaining contributions both inside and outside the subspace. As highlighted in Figure 1, both the bulk and dominant subspaces capture significant gradient energy; our methods are designed to exploit both.

**Subspace Adjustment.** GrassWalk and GrassJump, differ only in how they adjust the subspace. In GrassWalk, we first initialize the subspace from the initial gradient matrix, $G_0$, using a rank-$r$ SVD as in equation 2. The subspace is subsequently updated every $T$ iterations, which we refer to as the subspace update interval. To update, we employ the exponential map on the Grassmannian manifold (Bendokat et al., 2024), moving in a random direction from the current subspace. Concretely, we sample a random matrix $\mathbb{X} \in \mathbb{R}^{m \times r}$ to define the update direction in the tangent space. The update rule in equation 4 requires the SVD of $\mathbb{X}$ to move along the corresponding geodesic. Since we employ random directions, we approximate this decomposition using randomized SVD to reduce computational cost, denoting the result as $\widehat{U}_X \widehat{\Sigma}_X \widehat{V}_X^\top$. Here, $\eta$ denotes the update step size.

$$S_{t+1}(\eta) = (S_t \widehat{V}_X \quad \widehat{U}_X) \begin{pmatrix} \cos \widehat{\Sigma}_X \eta \\ \sin \widehat{\Sigma}_X \eta \end{pmatrix} \widehat{V}_X^\top + S_t (I - \widehat{V}_X \widehat{V}_X^\top) \tag{4}$$

For GrassJump, we adopt fully random projection matrices, effectively jumping from one point on the Grassmannian to another every $T$ iterations. At each update, we generate a random orthonormal matrix by applying QR decomposition to a randomly sampled matrix. This approach yields fine-grained random projections, in contrast to block-wise or column-wise subspace selection.

**Informing the Optimizer of Subspace Updates.** A key factor in the success of low-rank gradient methods is properly adapting the optimizer states when the underlying gradient subspace changes. This adaptation becomes especially critical when leveraging the optimizer's history to recover information lost during projection, as we discuss next.

In Adam, the momentum update rules compute weighted averages of the first- and second-order gradient moments using the parameters $\beta_1$ and $\beta_2$, as shown in the following equations.

$$M_t \leftarrow \beta_1 \cdot M_{t-1} + (1 - \beta_1) \cdot \widetilde{G}_t \tag{5}$$

$$\mathcal{V}_t \leftarrow \beta_2 \cdot \mathcal{V}_{t-1} + (1 - \beta_2) \cdot \widetilde{G}_t^2 \tag{6}$$

When the subspace is updated, we rotate Adam's moments onto the new basis so that the optimizer remains aligned with the updated subspace. Orthogonal projection works well for the first moment but not for the second, since Adam involves nonlinear operations. To handle this, we treat Adam's states as statistical estimates of the first and second moments of each gradient coordinate, and thus using equation 7 and equation 8 for our adaptive optimizer (AO). A similar perspective has also been adopted in prior state-of-the-art methods (Robert et al., 2025; Anonymous, 2025).

$$M_t \leftarrow \beta_1 (S_t^\top S_{t-1} M_{t-1}) + (1 - \beta_1) \widetilde{G}_t \tag{7}$$

---

**Algorithm 1** GrassWalk , GrassJump

---

**Require:** $W_t$, $G_t \in \mathbb{R}^{m \times n}$ with $m \leq n$ (w.l.o.g.), learning rate $\alpha$, decay rates $\beta_1$ and $\beta_2$, subspace update interval $T$, recovery scaling limiter factor $\zeta$.

    $S_0 \leftarrow U[:, :r]$, where $U, S, V \leftarrow \text{SVD}(G_0)$

    **while** Convergence **do**

        **if** *step* mod $T == 0$ **then**

            $S_t \leftarrow$ random rank-$r$ orthonormal matrix

            generate a random matrix; update the subspace in its direction as per equation 4

            calculate low-rank gradient $\widetilde{G}_t = S_t^\top G_t$

            adaptive optimizer as in equation 7 and equation 8

        **else**

            $S_t = S_{t-1}$

            regular adam as in equation 5 and equation 6

        **end if**

        $\widehat{G}_t^O \leftarrow$ optimizer's output, $\widehat{G}_t = S_t \widetilde{G}_t^O$

        compute $\Lambda_t$ as in equation 9

        $W_t \leftarrow W_{t-1} - \alpha \cdot \widehat{G}_t - \alpha \cdot \Lambda_t$

    **end while**

---

$$V_t \leftarrow \beta_2 \left[ (1 - \beta_2^{t-1}) |(S_t^\top S_{t-1})^2 \cdot (V_{t-1} - M_{t-1}^2) + (S_t^\top S_{t-1} \cdot M_{t-1})^2| \right] + (1 - \beta_2) \widetilde{G}_t^2. \tag{8}$$

**Recovering Information Lost in Low-Rank Projections.** The low-rank projection discards the residual $\Delta_t = G_t - S_t \widetilde{G}_t$ when mapping the gradient into a lower-dimensional subspace. Based on the observation that the scale ratio between dominant and bulk subspaces is consistent (Zhu et al., 2025; Chen et al., 2025b), we reintroduce this signal by columnwise rescaling of $\Delta_t$ according to the ratio between the optimizer's output $\widetilde{G}_t^O$ and the raw low-rank gradient $\widetilde{G}_t$, as shown in equation 9. This enables the use of stateful optimizer dynamics without storing the full optimizer states. With a growth-rate limiter $\zeta$, we prevent the scaling from diverging. Specifically, if $\|\Lambda_t\|/\|\Lambda_{t-1}\| > \zeta$, we rescale as per equation 10.

$$\phi_t(G_t)_i = \frac{\|\tilde{G}_{t,:,i}^O\|}{\|\tilde{G}_{t,:,i}\|}, \qquad \Lambda_t = \phi_t(G_t)\,\Delta_t, \tag{9}$$

$$\Lambda_t \leftarrow \Lambda_t \cdot \frac{\zeta \|\Lambda_{t-1}\|}{\|\Lambda_t\|}. \tag{10}$$

Several works (Anonymous, 2025; Zmushko et al., 2025; Chen et al., 2025b; Zhu et al., 2025; Robert et al., 2025) have employed various recovery scaling (RS) techniques, and we found this structure to be the most effective complement to our fine-grained random projection matrices. The final weight update is then expressed as:

$$W_t \leftarrow W_{t-1} - \alpha \hat{G}_t - \alpha \Lambda_t. \tag{11}$$

The pseudo code of GrassWalk and GrassJump can be found in Algorithm 1.

## 5 RESULTS AND ABLATIONS

**Systematic Ablation.** We ablate the role of the three discussed components by considering four baseline subspace update rules: **(a) Grassmannian subspace tracking** from SubTrack++ (Anonymous, 2025) that tracks the subspace by minimizing a projection error and updating along a Grassmannian geodesic. From the estimation error, they form a tangent vector to find the optimum subspace; see their paper for further details. **(b) Random walk on Grassmannian** which is the subspace update rule of GrassWalk as per equation 4; **(c) Random projections**, which recompute a fresh orthonormal basis at each update and is used in GrassJump ; and **(d) SVD-based updates** as in GaLore (Zhao et al., 2024a). To isolate the effect of each component, we subsequently incorporate AO and RS both individually and jointly into each baseline, and report the resulting evaluation loss under matched training and evaluation conditions across all configurations. The experiments are performed on the pre-training of a Llama-1B architecture using the C4 dataset.

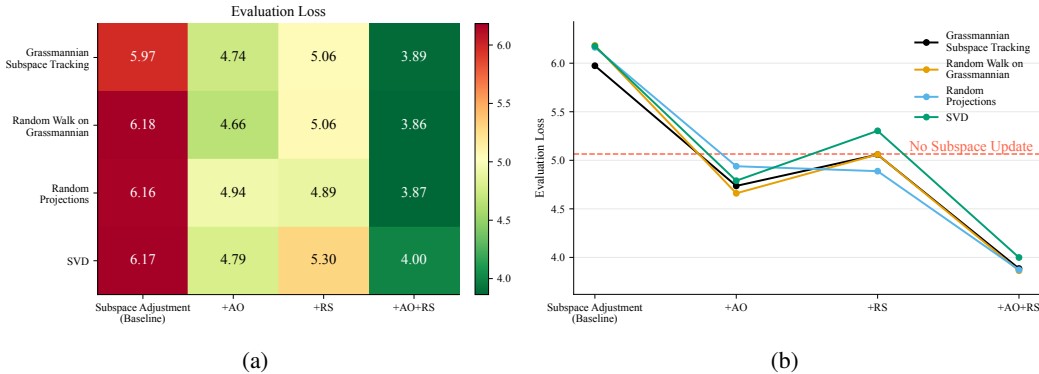

Figure 3: We ablate (i) the subspace update method: Grassmannian tracking, Grassmannian random walk, random projections, and SVD; (ii) adaptive-optimizer (AO), and (iii) recovery scaling (RS), reporting evaluation loss (lower is better). The "No Subspace Update" variant freezes the initial SVD subspace $S_0$; because the subspace is fixed, AO is inapplicable and only RS is active.

As shown in Figure 3, Grassmannian subspace tracking achieves the lowest loss among the update rules when neither AO nor RS is applied, demonstrating the effectiveness of structured manifold-based updates, particularly in comparison to SVD. This result further underscores the limitations of SVD, which is known to be sensitive to noise (He et al., 2025) and to discard prior information (Anonymous, 2025), yielding performance comparable to our random subspace updates.

Adding AO to subspace adjustment strategies yields the largest improvement in nearly all settings, with the notable exception of **random projections**. We attribute this to the extent to which each method preserves the "core" gradient subspace. Grassmannian updates modify the previously learned subspace through controlled rank-1 rotations; whether it is subspace tracing or a random walk on the manifold, the divergence remains small. Similarly, SVD explicitly captures dominant directions; although it is susceptible to noise, the resulting projection still retains most of the informative components. In contrast, random projections select arbitrary subspaces that may discard salient signal. Consequently, RS plays a more critical role in this setting, as the discarded information is more likely to be essential, making its recovery significantly more beneficial. By contrast, the weaker performance of RS without AO is expected. RS relies on the scale factors computed by the Adam optimizer; however, if the optimizer is not informed of subspace changes, these scales are corrupted by outdated bases and fail to reflect the intended columnwise rescaling.

In a complementary setting, we freeze the initial subspace $S_0$, obtained via an SVD of the first gradient matrix at the start of training. With the subspace fixed, AO remains inactive and only RS is applied. As shown in Figure 3, this frozen variant performs comparably to Grassmannian updates + RS, suggesting that the core subspace is largely captured from the very first iteration and can yield competitive performance once the lost information from low-rank projection is recovered. Nevertheless, because the true subspace inherently evolves during training, enabling all components continues to provide substantial gains across all subspace adjustment methods.

The results reveal an important finding: when lost information is effectively recovered and the optimizer is informed of subspace changes, random projections can outperform other subspace update methods. Thus, they are not merely an efficient substitute for more expensive approaches; owing to the relatively flat curvature of the gradient space during training, random projections can also aid in escaping local minima and act as a form of systematic regularization in the optimization process.

We have included additional ablations for investigating the sensitiviy of GrassWalk and GrassJump to different hyperparameters, which are reported at Appendix B.

**Pre-Taining Experiments.** We evaluate multiple baselines to compare them during LLaMA-1B and Qwen-1.5B pretraining for 10K steps under identical settings. All experiments are conducted on a single A6000 GPU, and we report the final evaluation loss. Method-specific parameters are set according to reported configurations in the original paper, with the subspace updated every 100 steps. Hyperparameters are reported in Appendix D.

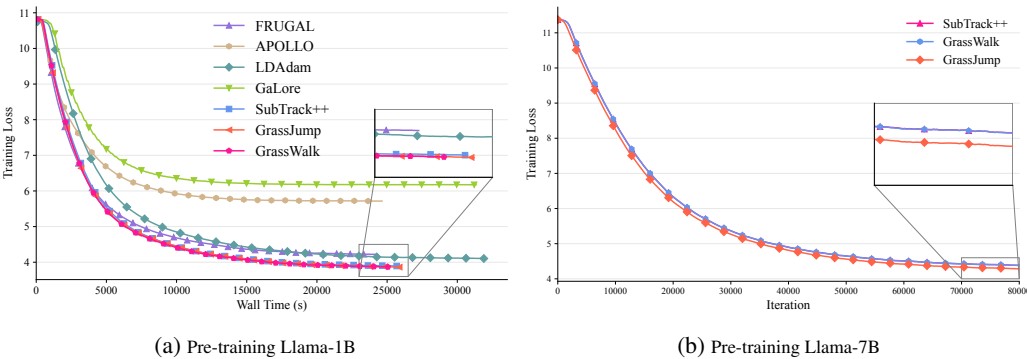

(a) Pre-training Llama-1B  (b) Pre-training Llama-7B

Figure 4: Comparison of different methods on LLaMA pretraining. (a) Wall-clock training curves for LLaMA-1B across all baselines. (b) Pretraining results for LLaMA-7B across selected methods, excluding weaker baselines due to their large performance gap.

The results in Table 1 show that, while offering GaLore-level memory and low computational cost, GrassWalk and GrassJump achieve superior performance. As illustrated in Figure 4-a, the wall-clock time of our randomised methods matches that of the most computationally efficient methods, such as APOLLO (Zhu et al., 2025), FRUGAL (Zmushko et al., 2025) and SubTrack++ (Anonymous, 2025), demonstrating superior convergence compared with other methods.

Table 1: Comparison of low-rank methods on pretraining LLaMA-1B and Qwen 1.5B models. We report evaluation loss (↓), peak memory (GB), and wall-time (m). Best results are in **bold**, and second best are underlined.

| Arch. | Method | Eval. Loss | Peak Mem. (GB) | Wall Time (m) |
|---|---|---|---|---|
| LLaMA-1B | AdamW [Full-Rank] | 4.10 | 35.2 | 417.0 |
| | GaLore (Zhao et al., 2024a) | 6.17 | 31.1 | 522.2 |
| | APOLLO (Zhu et al., 2025) | 5.71 | 35.5 | 410.5 |
| | LDAdam (Robert et al., 2025) | 4.10 | 34.9 | 532.8 |
| | FRUGAL (Zmushko et al., 2025) | 4.22 | 39.3 | 405.1 |
| | SubTrack++ (Anonymous, 2025) | 3.89 | 32.6 | 429.2 |
| | GrassWalk [Ours] | **3.86** | 32.0 | 418.6 |
| | GrassJump [Ours] | 3.87 | 32.1 | 432.5 |
| | GrassJump - No QR [Ours] | 3.87 | 32.1 | 415.2 |
| Qwen-1.5B | AdamW [Full-Rank] | 4.84 | 37.7 | 421.0 |
| | SubTrack++ (Anonymous, 2025) | 4.70 | 33.1 | 436.4 |
| | GrassWalk [Ours] | 4.68 | 33.6 | 436.6 |
| | GrassJump [Ours] | **4.66** | 33.1 | 437.0 |

To assess the generalizability in larger models and for long-run trainings, we report the evaluation loss when pre-training a Llama-7B architecture up to 100K iterations (Table 2). In this experiments, GrassJump outperforms GrassWalk and other baselines, highlighting the importance of thoroughly exploring model parameters and applying regularization at scale. Here, we exclude other baselines, as their performance was consistently weaker. Also, the training dynamics and evaluation loss during training are attached in Appendix C, with hyperparameters reported in Appendix D.

## 6 RELATED WORKS

**Efficient training background.** LoRA (Hu et al., 2021) reduces fine-tuning memory via low-rank adapters, with extensions such as QLoRA (Dettmers et al., 2024) and Deep LoRA (Yaras et al., 2024) improving efficiency and robustness. Further variants enhance adaptation (Lialin et al., 2023; Renduchintala et al., 2024; Xia et al., 2024; Pan et al., 2024), while other approaches boost memory efficiency by compressing activations (Miles et al., 2024) or reformulating optimization via block

Table 2: Comparison of low-rank gradient methods for pretraining LLaMA-7B. We report evaluation loss (↓), peak memory usage (GB), and wall-clock time (hours) after 10k and 100k of training iterations. Other baselines are omitted as their performance differs substantially from these three methods. Best results are in **bold**.

| Method | Eval. Loss - 10k (Wall Time) | Eval. Loss - 100k (Wall Time) | Peak Mem. (GB) |
|---|---|---|---|
| SubTrack++ (Anonymous, 2025) | 4.37 (15.1 hours) | 3.36 (93.2 hours) | 49.4 |
| GrassWalk [Ours] | 4.37 (15.1 hours) | 3.37 (93.2 hours) | 49.4 |
| GrassJump [Ours] | **4.27** (14.9 hours) | **3.34** (92.6 hours) | 49.4 |

coordinate descent (Luo et al., 2024). FLora (Hao et al., 2024) provides a complementary perspective by showing that LoRA can be interpreted as a random projection–based gradient compressor. They resamples projection matrices to achieve effectively high-rank updates while maintaining sublinear optimizer state complexity. Another line of work exploits the structure of high-dimensional data by projecting it into evolving low-dimensional subspaces. Incremental and Grassmannian-based methods have been proposed for subspace tracking under partial observations (Balzano et al., 2011), noise (Zhang & Balzano, 2016; Kasai, 2017), and geodesic evolution (Blocker et al., 2023), offering a principled foundation for gradient projection techniques in LLM training.

**Low-rank gradient methods.** As optimizers like Adam (Kingma & Ba, 2017) account for a significant portion of memory, there are many methods (Modoranu et al., 2024), (Zhang et al., 2024) that aim to reduce optimizer states. MicroAdam (Modoranu et al., 2024) compresses gradients with feedback correction, while Adam-mini (Zhang et al., 2024) partitions models into blocks with shared learning rates. (Gur-Ari et al., 2018), (Schneider et al., 2024) show that a substantial portion of gradients lies within a largely consistent subspace. GaLore (Zhao et al., 2024a) first leverage this fact to reduce optimizer's memory by projecting gradients onto a low-rank subspace, yielding large memory savings. Jaiswal et al. (2024) fine-tune only layers with low-dimensional gradient subspaces, while Grass (Muhamed et al., 2024) saves memory via sparse gradient projections. Ramesh et al. (2024) achieve efficiency by dynamically updating only a subset of parameters. GoLore (He et al., 2025) addresses GaLore's convergence issue and by injecting random projections in later iterations, ensures convergence. Fira (Chen et al., 2025b) uses norm-based scaling to transfer the adaptive behavior of a low-rank optimizer to full-rank updates while performing SVD for subspace update.

APOLLO (Zhu et al., 2025) approximates channel-wise scaling using an auxiliary random low-rank space, effectively coarsening learning-rate adaptation with SGD-like memory. The RSO framework (Chen et al., 2025c) decomposes training into sequences of randomized lower-dimensional subproblems. Adapprox (Zhao et al., 2024b) targets Adam's second moment with randomized low-rank approximations, adaptive rank and similarity guidance. GreedyLore (Chen et al., 2025a) explores greedy low-rank gradient compression with error-feedback and semi-lazy subspace updates. Also, FRUGAL(Zmushko et al., 2025) leverages gradient splitting: it applies stateful updates in a low-dimensional space and state-free methods (SGD/signSGD) (Bernstein et al., 2018) along remaining directions, using columnwise random projections.

# 7 DISCUSSION AND CONCLUSION

Our analysis showed that while a low-rank core subspace captures substantial gradient energy, its dominance fades over time and across deeper layers, and that the subspace evolves within a nearly flat curvature. These findings explain both the strengths and weaknesses of existing randomized and structured approaches, and motivate our proposed algorithms. By incorporating subspace-aware random walks and jumps, together with optimizer adaptation and recovery mechanisms, GrassWalk and GrassJump consistently outperform prior state-of-the-art methods, while retaining GaLore-level memory efficiency. The results demonstrate that randomized strategies are not merely computational shortcuts: when designed with awareness of gradient dynamics, they can become principled tools for stability, generalization, and convergence speed. More broadly, our study reframes randomization as a feature of low-rank gradient training, and highlights the importance of investigating the training dynamics. We hope our work helps establish a stronger foundation for the next generation of memory-efficient optimization methods.

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

## A    CORE SUBSPACE ENERGY VS. RANK

In Figure 1, we illustrate the fraction of gradient energy captured by the core subspace, as defined in equation 3. These plots were generated during pre-training of a Llama-1B architecture using a subspace rank of $r = 512$. To further examine this phenomenon across different subspace capacities, we additionally include results for ranks 256 (Figure 5) and 1024 (Figure 6).

As anticipated, increasing the rank also increases the energy fraction. Crucially, even with rank 1024, a considerably high rank for a 1B model, the fractions stabilize between 70% and 80% after initial iterations (with a significant 20% to 30% scattered on the orthogonal space), consistently demonstrating the decay pattern across nearly all layer types, except for MLP-down projection. For rank 256, the fractions are significantly lower, settling around 40% to 50%. This extended experimentation strongly validates the conclusion drawn from Figure 1.

## B    ADDITIONAL ABLATIONS

In this section we report the result of ablations against different hyperparameters.

### B.1    SUBSPACE RANK

The rank of the core subspace, when pre-training the Llama-1B architecture with a subspace update interval of 200, impacts the evaluation loss as shown in the Table 3. While a higher rank generally yields better loss, the marginal benefit is limited. Notably, in the GrassJump, the reduced dependency on a specific subspace allows for effective performance even with significantly lower ranks.

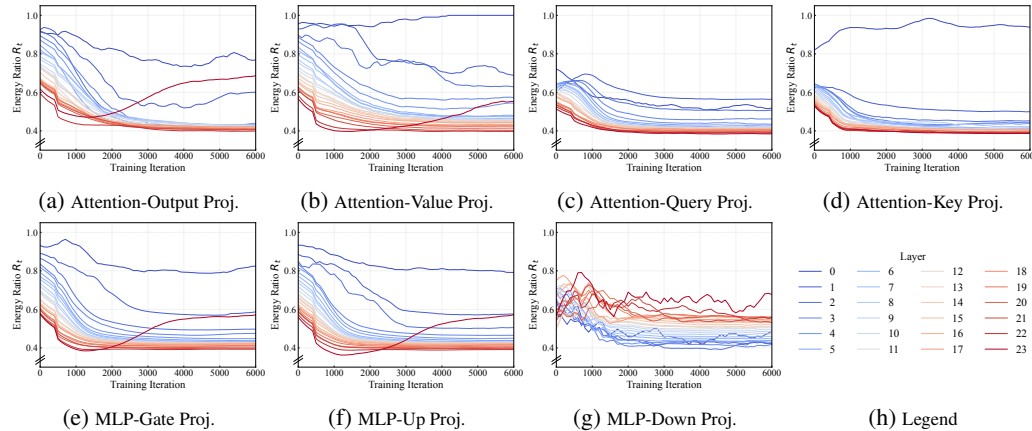

Figure 5: Each decoder layer stack includes seven layer types in the Llama-1B model. The plots show the fraction of gradient-matrix energy explained by a rank 256 approximation. Despite a high lower bound for this rank, this fraction declines over training, and deeper layers generally exhibit smaller fractions.

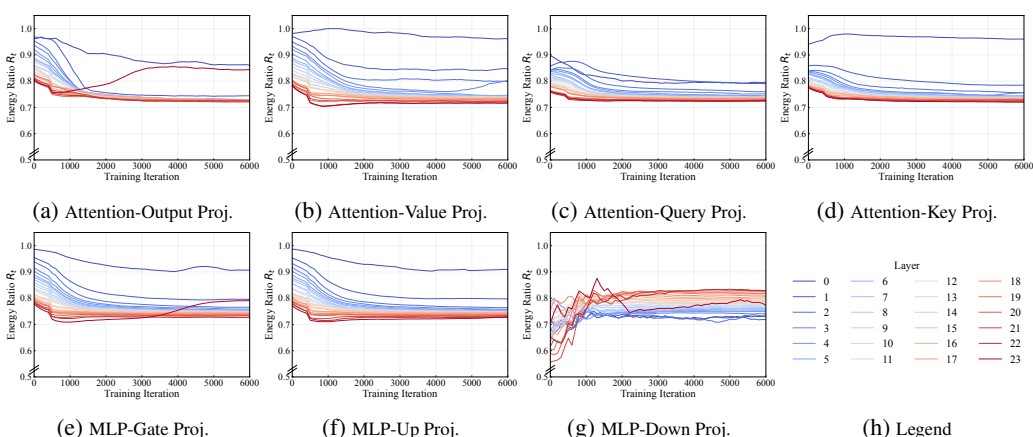

Figure 6: Each decoder layer stack includes seven layer types in the Llama-1B model. The plots show the fraction of gradient-matrix energy explained by a rank 1024 approximation. Despite large rank for this model size and a high lower bound, this fraction declines over training, and deeper layers generally exhibit smaller fractions.

Table 3: Final evaluation loss (↓) after pre-training a Llama-1B architecture for 10k iterations and with different subspace ranks.

| Method | r = 256 | r = 512 | r = 1024 |
|---|---|---|---|
| GrassWalk | 4.62 | 4.52 | 4.43 |
| GrassJump | 4.54 | 4.50 | 4.45 |

## B.2 SUBSPACE UPDATE INTERVAL

Table 4 summarizes the impact of update frequency on pre-training of aLlama-1B architecture with rank 512. Notably, no subspace update results in a final evaluation loss of 5.06 (Figure 3), underscoring the necessity of subspace adaptation. Crucially, excessively frequent GrassJump updates degrade performance, suggesting that drastic subspace changes disrupt the optimizer's effectiveness and necessitate sufficient iteration for convergence.

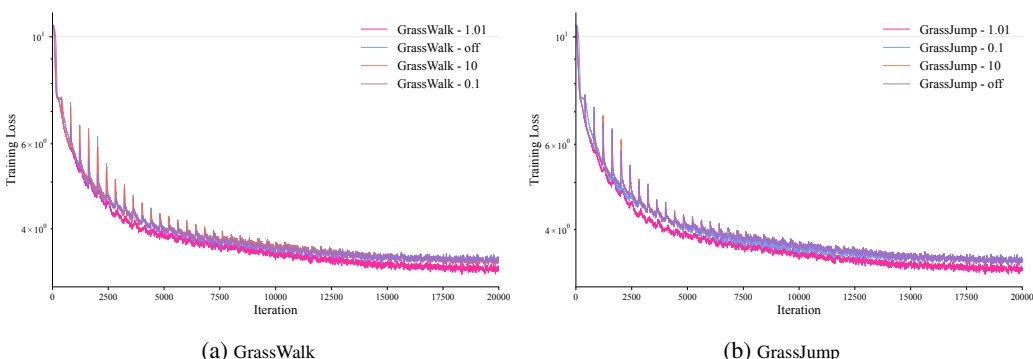

(a) GrassWalk

(b) GrassJump

Figure 7: Training dynamics of pre-training a Llama-1B architecture with rank 512 for 10k iterations using different values for norm-growth limiter $\zeta$.

## B.3 NORM-GROWTH LIMITER

In Figure 7, we examine the effect of different norm-growth limiter values on the training dynamics of a Llama-1B model trained with rank 512 using the GrassWalk and GrassJump methods. As shown, this parameter acts similarly to a gradient-clipping coefficient, regulating training spikes. Furthermore, Table 5 demonstrates that mitigating these spikes, while avoiding excessive suppression of the recovered signal (e.g., ($\zeta = 0.1$)) has a substantial impact on the final evaluation loss.

Table 4: Final evaluation loss ($\downarrow$) after pre-training a Llama-1B architecture for 10k iterations with $r = 512$ and with different subspace update intervals.

| Method | T = 20 | T = 50 | T = 100 | T = 200 | T = 500 | T = 1K | T = 2K |
|---|---|---|---|---|---|---|---|
| SubTrack++ (Anonymous, 2025) | NC | 3.86 | 3.89 | 3.99 | 4.25 | 4.45 | 4.68 |
| GrassWalk [Ours] | 3.72 | 3.79 | 3.86 | 3.98 | 4.25 | 4.46 | 4.68 |
| GrassJump [Ours] | 4.33 | 4.34 | 3.87 | 3.96 | 4.15 | 4.75 | 4.89 |

Table 5: Final evaluation loss ($\downarrow$) after pre-training a Llama-1B architecture for 10k iterations with $r = 512$ and with different values for the norm-growth limiter $\zeta$.

| Method | $\zeta = 0.1$ | $\zeta = 1.01$ | $\zeta = 10$ | $\zeta = \infty$ |
|---|---|---|---|---|
| GrassWalk | 3.44 | 3.26 | 3.41 | 3.42 |
| GrassJump | 3.45 | 3.30 | 3.44 | 3.44 |

## B.4 THE STEP-SIZE OF THE GRASSMANNIAN UPDATES

GrassJump employs purely random subspace selection for maximal exploration. Conversely, Grass-Walk incorporates an $\eta$ parameter, representing the step-size for updates on the Grassmannian manifold that can affect its performance. Ablation studies indicate that the step-size is inconsequential to the final pre-training evaluation loss of GrassWalk on pre-training a Llama-1B model, as demonstrated in Table 6. Notably, the step-size's appearance in the sine and cosine terms of equation 4 inherently leads to oscillatory behavior after scaling.

Table 6: Final evaluation loss ($\downarrow$) after pre-training a Llama-1B architecture for 10k iterations with $r = 512$ using GrassWalk with diffent subspace update step-size $\eta$.

| Method | $\eta = 10$ | $\eta = 100$ | $\eta = 1000$ | $\eta = 10000$ |
|---|---|---|---|---|
| GrassWalk | 3.9812 | 3.9786 | 3.9815 | 3.9890 |

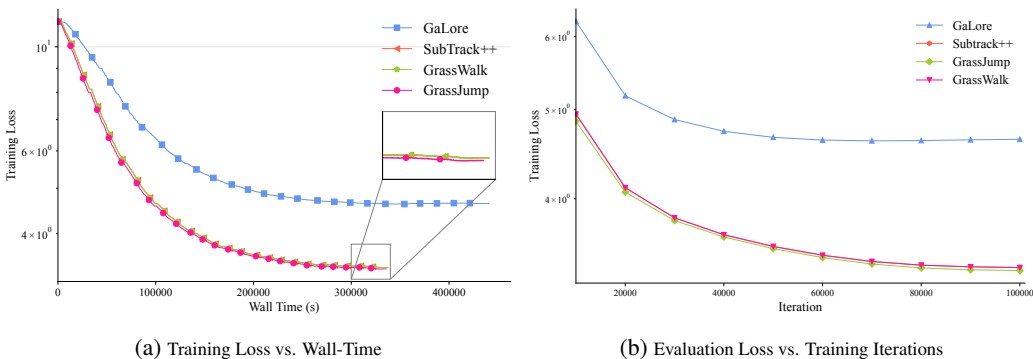

(a) Training Loss vs. Wall-Time          (b) Evaluation Loss vs. Training Iterations

Figure 8: Training and evaluation dynamics of pre-training a Llama-7B architecture with rank 1024 for 100k iterations across different methods.

## C  LONG-RUN TRAINING OF 7B MODEL

We pre-trained the Llama-7B architecture using GrassWalk, GrassJump, and SubTrack++ (Anonymous, 2025) for 100K steps. The results are presented in Table 2, and Figure 8 shows GrassJump consistently outperforms both in training dynamics and evaluation loss. Notably, GrassWalk achieves comparable performance without requiring gradient tracking.

## D  PRE-TRAINING HYPERPARAMETERS

The hyperparameters of the experiments are reported in Table 7. All experiments are conducted on A6000 GPUs.

Table 7: Hyperparameters of pre-training Llama-based and Qwen architectures.

| | | Llama-1B | Llama-7B | Qwen-1.5B |
|---|---|---|---|---|
| Architectural Parameters | Hidden | 2048 | 4096 | 1536 |
| | Intermediate | 5461 | 11008 | 8960 |
| | Heads | 24 | 32 | 12 |
| | Layers | 32 | 32 | 28 |
| Shared Parameters | Learning Rate | 1e-4 | 1e-4 | 1e-4 |
| | Batch Size | 32 | 8 | 16 |
| | Gradient Accumulation | 2 | 4 | 4 |
| | Iterations | 10k | 100k | 10k |
| | Gradient Clipping | | 1.0 | |
| | Warmup Steps | | 1000 | |
| | scale | | 0.25 | |
| | dtype | | bfloat16 | |
| Low-Rank Optimizer Methods Parameters | Rank | 512 | 1024 | 256 |
| | Subspace Update Interval | 100 | 500 | 100 |
| | Step-Size | | 10000 | |

## E  FLAT CURVATURE

Figure 2 shows the distribution of the singular values of the tangent vectors of the subspace-estimation error across layers and training iterations. Figure 9 plots the Frobenius norm of these singular values for each layer over the course of training. As illustrated, nearly all layers exhibit extremely small values, indicating that the subspace-estimation error has very low sensitivity in almost every direction.

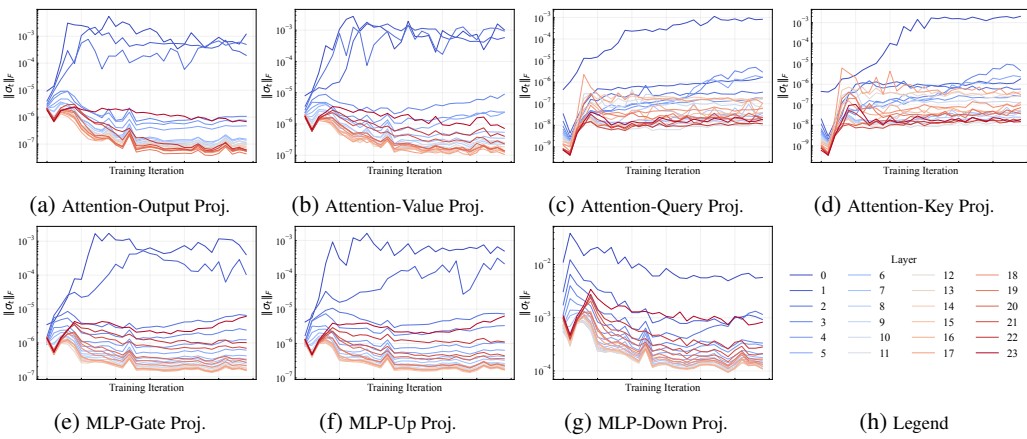

Figure 9: Frobenius norm of singular values of the subspace estimation error derivative across different projection layers in the LLaMA-1B architecture. Each plot shows the norm within a given layer type (aggregated across all 24 decoder layers) as training progresses. Almost all the layers demonstrate extremely small values, indicating the flat curvature of the gradient subspace optimization space.

