# OpenReview forum: "Randomized Gradient Subspaces for Efficient Large Language Model Training"
_ICLR.cc/2026/Conference — Submitted to ICLR 2026_

### Official Review · Reviewer_VgBp · 2025-10-27

**Soundness:** 3
**Presentation:** 2
**Contribution:** 2
**Rating:** 4
**Confidence:** 3

**Summary:**

This paper addresses the high memory demands of training LLMs, which are often bottlenecked by optimizer states. The authors analyze gradient subspace dynamics, finding that the dominance of a low-rank subspace declines over time, especially in deeper layers, and evolves in a nearly flat curvature. Motivated by these insights, they introduce GrassWalk and GrassJump, two randomized methods that update the gradient subspace using random walks and jumps on the Grassmannian manifold3333. By adapting the optimizer to these changes and recovering information lost during projection, the methods achieve superior performance and memory efficiency in pretraining LLaMA-1B and LLaMA-7B models4.

**Strengths:**

- The paper's primary strength lies in its analysis of gradient subspace dynamics (Section 3). The finding that the dominance of the low-rank "core" subspace diminishes over training and in deeper layers (Figure 1) is a significant and non-obvious contribution. It provides a strong "why" for moving beyond simple subspace tracking.
- The analysis of the subspace estimation error (Figure 2) provides a principled, geometric justification for using randomized methods. The conclusion of a "nearly flat curvature" compellingly reframes random projections from a mere computational shortcut to a motivated strategy for escaping poor local optima in the subspace manifold.
- The proposed methods GrassWalk and GrassJump claim to achieve state-of-the-art results on LLaMA-1B and 7B pre-training (Tables 1 & 2). Outperforming strong, contemporary baselines like SubTrack++ in final evaluation loss while maintaining equivalent memory efficiency is a clear and impressive empirical win.
- The ablation in Figure 3 dissects the method into its three key components (Subspace Update, Adaptive Optimizer, Recovery Scaling) and clearly demonstrates that the randomized update strategies are only effective when combined with both AO and RS. This transparency is crucial for understanding the method's mechanics.

**Weaknesses:**

- The two components that appear to drive the majority of the performance gain (adaptive optimizer and recovery scaling) are explicitly and fairly acknowledged as being adapted from prior work (Robert et al., 2025; Anonymous, 2025; Chen et al., 2025b). The core algorithmic novelty is limited to the randomized subspace update rule itself. This makes the paper feel more like a good analysis that validates a simple, effective tweak to an existing SOTA framework (e.g., SubTrack++).
- The paper introduces GrassWalk as a more "principled" random walk on the manifold, but GrassJump (a full, "unprincipled" random projection) performs better on the larger 7B model. This finding somewhat undermines the motivation for the more complex GrassWalk method and suggests the benefits may just stem from the regularization of random projections, rather than a controlled "walk."
- The intuitive  "flat curvature" argument from Figure 2 could be more formally defined. Is this the curvature of the main loss landscape $L(W)$, or the curvature of the auxiliary objective $f(S) = ||G_t - S S^\top G_t||_F^2$ on the Grassmannian manifold $Gr(r, n)$? Clarifying this would strengthen the paper's central theoretical motivation.

**Questions:**

- The performance of GrassWalk and GrassJump likely depends heavily on how frequently the subspace is updated ($T$) and how far each update moves on the Grassmannian ($\eta$). Could you please provide sensitivity plots or an empirical study showing how evaluation loss varies with them? In particular, does GrassWalk’s random-walk dynamics require fine-tuning of to maintain numerical stability?
- RS rescales the discarded gradient component using optimizer outputs. However, when random projections drastically change the subspace, the discarded portion $\Delta_t$ may be nearly uncorrelated with prior directions, raising the risk that RS amplifies noise rather than useful signal. Could you analyze or visualize this effect—for instance, by plotting the ratio $\|\Lambda_t\|/\|G_t\|$ over training, or showing how performance changes when RS is disabled for random projections? Clarifying whether RS stabilizes or destabilizes random subspace updates would help interpret Figure 3.
- Since GrassWalk introduces gradual manifold exploration and GrassJump performs abrupt rerandomization, they seem complementary. Did the authors consider hybrid schedules, e.g., performing several GrassWalk steps followed by an occasional GrassJump to escape flat regions? This could potentially balance stability and exploration.

---

> ### Author Response · Authors · 2025-11-22
>
> We thank the reviewer for their detailed review, and for confirming our contribution significant and non-obvious. We have addressed the raised concerns and questions below, and have applied the reviewer’s suggestions into our rebuttal revision to strengthen the paper.
>
> > **W1. Comparison with Other Baselines**
>
> Our work introduces a paradigm shift in low-rank optimization by challenging the **core assumption** of subspace tracking and adjustment, employed in many prior works, from GaLore [1], to Fira [2], SubTrack++ [3], LDAdam [4], and others [5]. While we share the mathematical foundation (Grassmannian manifolds) of SubTrack++ [3], our contribution lies in **revisiting this core assumption**.
>
> Our empirical analysis of gradient energy and gradient subspaces’ curvature shows that:
> - A substantial fraction of gradient energy consistently lies outside the tracked subspace (Figure 1).
> - The projection error is not sensitive to subspace updates, suggesting that the curvature is flat and precise subspace tracking may not work (Figures 2 and 9).
>
> These observations motivate the use of randomized projections, and we show that, within GrassWalk and GrassJump optimization regimes, random steps or jumps can perform competitively with sophisticated subspace-tracking updates, and even outperforming them (Tables 1 and 2).
>
> We also investigated the effect of **AO** and **RS** components on top of **various subspace selection methods** (Figure 3), empirically validating that these, along our randomized subspace changes, are the superior algorithm design.
>
> > **W2. GrassWalk vs. GrassJump**
>
> We agree that GrassJump’s strong performance raises questions about where principled walks matter. We clarify our intended distinction:
>
> Our analyses motivate the use of random projections for their two key benefits:
>
> **1) Exploration:** The significant energy in the orthogonal space implies crucial information is often overlooked. Random projections act as a forcing exploration beyond dominant features.
>
> **2) Generalization:** In a flat curvature, random projections facilitate escaping local minima traps, thereby improving the generalization.
>
> The GrassJump and GrassWalk are inspired by these benefits. GrassJump utilizes sudden subspace changes for aggressive exploration. In contrast, GrassWalk employs controlled, rank-1 random steps to escape local minima without sudden changes.
>
> These distinct properties offer different advantages; GrassWalk shows greater robustness to frequent updates (Table 4) as the adjustments are more controlled, whereas GrassJump performs better with lower ranks (Table 3) due to their independence to a core subspace. In addition, per reviewer suggestion, we have explored a hybrid approach, explained in our responses to Q3.
>
> > **W3. Formal Definition of the Flat Curvature**
>
> The flatness corresponds to the objective on the Grassmannian manifold, suggesting that the projection error is not sensitive to the direction of subspace updates. We will make sure to reflect and clarify this point on the rebuttal edition.
>
> > **Q1. Subspace Update Interval and Step-Size Ablations**
>
> We have performed the requested ablation studies and incorporated the results to further illustrate our approach.
>
> ### **Update interval T (Table 4; rebuttal revision):**
> While increased update frequency generally improves final performance, overly frequent GrassJump updates degrade results. This suggests that drastic subspace changes disrupt optimizer convergence, requiring sufficient iteration time.
>
> *Table 1. Subspace update interval ablation*
> ||20|50|100|200|500|1000|2000|
> |:-|:-|:-|:-|:-|:-|:-|:-|
> |GrassWalk|3.72|3.79|3.86|3.98|4.25|4.46|4.68|
> |GrassJump |4.33|4.34|3.87|3.96|4.15|4.75|4.89|
>
> ### **Step-Size on the Grassmannian Manifold (Table 6; rebuttal revision):**
> GrassJump employs purely random subspace selection. Conversely, GrassWalk incorporates an η parameter, representing the step-size for updates on the Grassmannian Manifold. Ablation studies indicate that the step-size is inconsequential to the final pre-training evaluation loss of GrassWalk on a Llama-1B model. Notably, the step-size's appearance in the sine and cosine terms of equation 4 inherently leads to oscillatory behavior after scaling.
>
> *Table 2. Subspace Update Step-Size Ablation*
> ||10|100|1000|10000|
> |:-|:-|:-|:-|:-|
> |GrassWalk|3.9812|3.9786|3.9815|3.9890|
>
> > **Q2. The Role of Recovery Scaling Component in Random Projection Settings**
>
> We have shown performance changes in absence of RS with random projections in **Figure 3**. As described in lines 353 to 357 of the paper: “...random projections select arbitrary subspaces that may discard salient signals. Consequently, RS plays a more critical role in this setting, as the discarded information is more likely to be essential, making its recovery significantly more beneficial.”
> Also, the **column-wise** scaling of the RS component provides the essential granularity that prevents the dominance of noise effects [2, 6].

---

> ### Author Response · Authors · 2025-11-22
>
> > **Q3. Hybrid Approach: GrassWalk + GrassJump**
>
> Following the reviewer’s suggestion, we evaluated a simple schedule where GrassJump is applied with a probability $p$, for a LLaMA-1B model (rank 512). As demonstrated, the results are on-par, and with slight improving over either method alone.
>
> *Table 3. Hybrid Approach Ablation. $p$ shows the probability of performing GrassJump.*
> |Method|$p$|Eval. Loss|
> |:-|:-|:-|
> |Hybrid|0.3|3.85|
> |Hybrid|0.5|3.86|
> |GrassJump|1.0|3.87|
> |GrassWalk|0.0|3.86|
>
> ------------------------
>
> We thank the reviewer for their thoughtful feedback. We have incorporated all requested experiments and clarifications into the revised rebuttal version. We hope that these additions address the reviewer’s concerns, and we would be grateful if the reviewer could consider updating their evaluation accordingly.
>
> ------------------------
>
> [1] Zhao et al., 2024. *GaLore: Memory-Efficient LLM Training by Gradient Low-Rank Projection.*
>
> [2] Chen et al., 2024. *Fira: Can We Achieve Full-rank Training of LLMs Under Low-rank Constraint?*
>
> [3] Rajabi et al., 2025. *SubTrack++: Gradient Subspace Tracking for Scalable LLM Training.*
>
> [4] Robert et al., 2025. *LDAdam: Adaptive Optimization from Low-Dimensional Gradient Statistics.*
>
> [5] Liang et al., 2024. *Memory-Efficient LLM Training with Online Subspace Descent.*
>
> [6] Zhu et al., 2025. *APOLLO: SGD-like Memory, AdamW-level Performance.*

---

> ### Author Response · Authors · 2025-11-28
> **Follow-Up on Discussion**
>
> Dear Reviewer,
>
> We wanted to once again appreciate your detailed feedback. Also, if any of our earlier clarifications require further elaboration, or if additional concerns remain, we would be glad to provide more information while the discussion window is still open.
>
> Sincerely,
>
> Authors of paper 20846.

---

### Official Review · Reviewer_X2EV · 2025-10-31

**Soundness:** 2
**Presentation:** 2
**Contribution:** 2
**Rating:** 2
**Confidence:** 5

**Summary:**

This work introduces GrassWalk and GrassJump, two techniques used in the context of low-rank factorization of gradients to reduce the memory usage of optimizer states.

The paper builds on top of prior work SubTrack++, which introduces the idea of a "core" subspace.

The paper shows that at least 50% of the information is preserved in the core space, meaning there might be some useful information in the remaining dimensions that are not captured by the core space (non-core). They analyze the curvature of this non-core space and show it exhibits low curvature.

Their methods replace the rank-1 factorization of the tangent vector $\nabla F$ on the Grassmannian manifold in SubTrack++ with a random orthogonal matrix used as it is (called GrassJump) and alternatively inputting it into the exponential map on the Grassmannian manifold (called GrassWalk).

They test GrassWalk/Jump in pretraining settings for Llama models with 1B and 7B parameters and claim superior performance compared to the main baseline SubTrack++.

**Strengths:**

- Using random matrices to reduce the overhead of more accurate methods to perform low-rank decomposition, such as SVD, is an interesting idea
- the analysis in Section 3 suggests that the flatness of the non-core space (e.g. low-rank projection error, or the subspace that is not chosen for optimization) implies that random steps can be advantageous because the model will not be trapped in a sharp local minima

**Weaknesses:**

1. despite changing only the way the projection matrix $S_t$ is computed/updated, the paper has many paragraphs that are **almost identical** to the SubTrack++ paper in section 4. This suggests "thin slicing" or just a small increment over prior work.

2. using QR decomposition to orthogonalize the randomly generated matrix for GrassJump introduces a computational overhead as the runtime of QR-decomposition algorithm scales with $O(n^3)$

3. It seems like the methods introduced in this work are not much better than the rank-1 approximation of the tangent vector $\nabla F$ on the Grassmannian manifold. The results in Tables 1 and 2 are not significantly better compared to SubTrack++. As shown by the prior work in the area, a significantly lower running time is expected for the methods based on random/fixed projections.

**Questions:**

Questions:
1. lines 202-206: which rank did you use for Figure 1? is the rank correlated with the lower-bound of 50% that we see in Figure 1? Would Figure 1 look similarly (e.g. 50% as a lower bound) across multiple rank ratios? By rank ratio I mean rank divided by the dimension: e.g. for the original layer of size $(m, n)$ with $m < n$, this ratio is $r/m$.

2. How many seeds did you use to generate the results in Tables 1 and 2? Is the reported number an average across multiple runs? If yes, then it would be useful to see best, the works, the median, the mean and the standard deviation to understand how the result is affected by the randomness in the method.

3. About running times, related to Weakness 3: it seems like the QR decomposition in GrassJump is the root cause of the high running time of the method, while prior work [1] suggests much lower running time. Did you try to use low-rank orthogonal matrices without using QR decomposition? Note this is slow as it has to be run at each layer when you generate the random matrix.

4. What about choosing some random columns from the identity matrix instead of generating a random Gaussian matrix?


**Observations**:

1. I do not agree on the statement that at lines 11-12 in the abstract that *training LLMs is often bottlenecked by optimizer states which are dominating the footprint*. This is not true in general for the practical scale nowadays. When using Adam with a high model parallelism (e.g. many nodes to perform training across), the layers are split in such a way that the Adam states take a small fraction of the total memory (see Figure 1 in reference [2]). Your statement is true in low-memory settings, where we are forced to use batch-size 1 because otherwise the activations would require a lot of space. I would appreciate it if you could rephrase the statement in the abstract and/or add some more precise details, in order to target some specific settings fairly.

2. Please increase the labels on the x/y-axis of your plots, it is difficult to read them.



**Typos:**

- line 50: comma between **succeed** and **but**
- line 52: space after **GrassJump**
- line 53: **misleading → mislead** the optimizer (the very end of page 1)
- lines 230, 235, 236: space after **GrassWalk** and **GrassJump**



**References:**

[1] Authors: *Ionut-Vlad Modoranu, Mher Safaryan, Erik Schultheis, Max Ryabinin, Artem Chumachenko, Dan Alistarh*, Title: **FFT-based Dynamic Subspace Selection for Low-Rank Adaptive Optimization of Large Language Models**, https://arxiv.org/abs/2505.17967

[2] Authors: *Yiming Chen, Yuan Zhang, Yin Liu, Kun Yuan, Zaiwen Wen*, Title: **A Memory Efficient Randomized Subspace Optimization Method for Training Large Language Models**, https://arxiv.org/pdf/2502.07222

---

> ### Author Response · Authors · 2025-11-22
>
> We thank the reviewer for their detailed and constructive review, and for recognising the random approach interesting. We have addressed the raised concerns and questions below, and have applied the reviewer’s suggestions into our rebuttal revision to strengthen the paper.
>
> > **W1. Comparison with SubTrack++**
>
> Our work introduces a paradigm shift in low-rank optimization by challenging **the core assumption** of subspace tracking and adjustment, employed in many prior works, from GaLore [1], to Fira [2], SubTrack++ [3], LDAdam [4], and others [5]. While we share the mathematical foundation (Grassmannian manifolds) of SubTrack++ [3], our contribution lies in **revisiting this core assumption**.
>
> Our empirical analysis of gradient energy and gradient subspaces’ curvature shows that:
> - A substantial fraction of gradient energy consistently lies outside the tracked subspace (Figure 1).
> - The projection error is not sensitive to subspace updates, suggesting that the curvature is flat and precise subspace tracking may not work (Figures 2 and 9).
>
> These observations motivate the use of randomized projections, and we show that, within GrassWalk and GrassJump optimization regimes, random steps or jumps can perform competitively with sophisticated subspace-tracking updates, and even outperforming them (Tables 1 and 2).
>
> > **W2 and Q3. The Role of QR Decomposition and Its Runtime**
>
> While Gaussian projections are faster, in GrassJump, we perform a QR decomposition over them to derive a rank-$r$ orthonormal basis. This is standard practice for orthogonal random projections [6-9], ensuring orthonormality that will preserve norms, angles, and distributions [6-8].
>
> Per reviewer’s suggestion, we examined GrassJump without QR orthogonalization, which yielded on-par results in pre-training a 1B model (Table 1; rebuttal revision), reducing wall-time by **~3-4%**. This overhead shows that our QR operation poses marginal overhead (due to the low-frequency and low-rank nature of underlying matrix), and serves for theoretical stability.
>
> > **Q4. Random Sampling From Identity Matrix**
>
> Derived by our results, different random projections are likely to work as far as **1)** they have the enough coverage that a stateful optimizer like Adam can perform effectively **2)** we have a mechanism like recovery scaling to restore the lost information and **3)** we inform the optimizer when we change the subspace, so we don’t introduce noise or corruption to the computed states.
>
> Sparse projections are excellent for exploring a computationally cheaper alternative. However, we chose dense projection due to its superior preservation of the underlying structure [10], which is crucial for our optimizer's performance. We need higher accuracy as we also leverage this low-rank information for **recovery of the lost information** which can propagate errors. Given that our method introduces negligible overhead, we prioritized accuracy over efficiency in this work. Nonetheless, we acknowledge that investigating alternative projection methods presents a promising avenue for future research.
>
> > **W3. Performance Improvements**
>
> The improvements in gradient low-rank optimizers can be modest because several recent methods have shown great performance, match or outperform full-rank Adam [3, 4, 11]. Nonetheless, our results consistently show that both GrassWalk and GrassJump **match or exceed** the performance of SubTrack++ [3], LDAdam [4], and FRUGAL [11] across different architectures, ablations and training budgets (Tables 1 and 2, and Appendix B). These gains, while not large, are **consistent**, and support our main conclusion that **precise subspace tracking is not required** for strong performance.
>
> In addition, GrassWalk and GrassJump demonstrate on-par runtime with baselines like APOLLO [12], SubTrack++ [3], and FRUGAL [11] while being **~18%** faster than LDAdam [4] and GaLore [1]. In addition removal of QR decomposition from GrassJump, can further improve its computational efficiency.
>
> > **Q1. Figure 1 with Different Rank Ratios**
>
> We have added Figures 5 and 6 to the revision, showing results for ranks {256, 1024}. The conclusions remain consistent:
> - Rank 256 stabilizes around ~40–50% captured energy.
> - Rank 512 stabilizes around ~50–60%.
> - Rank 1024 stabilizes at ~70–80%, still leaving substantial energy in the orthogonal complement, especially given that this rank is considerably high for Llama-1B architecture.
>
> These results reinforce that the orthogonal space consistently carries nontrivial energy across a range of rank ratios, and almost all layers demonstrate the decay pattern, supporting our conclusion drawn from Figure 1.

---

> ### Author Response · Authors · 2025-11-22
>
> > **Q2. Experiments Seeds**
>
> We confirm the importance of multi-seed reporting in random methods. Thus, we have added multi-seed results for rank 512 in pretraining Llama-1B. The final version will include multi-seed reporting for all key experiments.
>
> Furthermore, the generalization and scability of our methods is also demonstrated across pre-training Llama-1B and Qwen-1.5B architectures (Table 1) and long-run Llama-7B (Table2), along with various ablations (Appendix B).
>
> *Table 1. Evaluation Loss of GrassWalk and GrassJump over Three Distinct Seeds.*
> |Method|Mean | std        |
> | :--------------- | :------------- | :--------- |
> | GrassWalk   | 3.865         | 0.003   |
> | GrassJump  | 3.879         | 0.007   |
>
> > **O1. Comparison with Parallel Methods**
>
> We appreciate the reviewer’s observation and have revised the abstract and introduction to specify the memory benefit in low-memory or single-node settings, where optimizer states can dominate footprint and restrict feasible batch sizes. Actually, algorithmic compression and parallelism are orthogonal and can help each other toward more efficient computations.
>
> >**O2. Labels and Typos**
>
> We have corrected all identified typos and increased the font size in Figures.
>
> ------------------------
>
>
> We thank the reviewer for their thoughtful feedback. We have incorporated all requested experiments and clarifications into the revised rebuttal version. We hope that these additions address the reviewer’s concerns, and we would be grateful if the reviewer could consider updating their evaluation accordingly.
>
>
> ------------------------
>
> [1] Zhao et al., 2024. *GaLore: Memory-Efficient LLM Training by Gradient Low-Rank Projection.*
>
> [2] Chen et al., 2024. *Fira: Can We Achieve Full-rank Training of LLMs Under Low-rank Constraint?*
>
> [3] Rajabi et al., 2025. *SubTrack++: Gradient Subspace Tracking for Scalable LLM Training.*
>
> [4] Robert et al., 2025. *LDAdam: Adaptive Optimization from Low-Dimensional Gradient Statistics.*
>
> [5] Liang et al., 2024. *Memory-Efficient LLM Training with Online Subspace Descent.*
>
> [6] Andrecut, 2024. *Residual Random Neural Networks.*
>
> [7] Banf and Hartwig, 2021. *Reasonable Effectiveness of Randomness in Scalable and Integrative Gene Regulatory Network Inference and Beyond.*
>
> [8] Murray et al., 2023. *Randomized Numerical Linear Algebra : A Perspective on the Field With an Eye to Software.*
>
> [9] Zhao et al., 2015. *Efficient Clustering on Riemannian Manifolds: A Kernelised Random Projection Approach.*
>
> [10] Chen et al., 2025. *Greedy Low-Rank Gradient Compression for Distributed Learning with Convergence Guarantees.*
>
> [11] Zmushko et al., 2024. *FRUGAL: Memory-Efficient Optimization by Reducing State Overhead for Scalable Training.*
>
> [12] Zhu et al., 2025. *APOLLO: SGD-like Memory, AdamW-level Performance.*

---

> ### Author Response · Authors · 2025-11-28
> **Follow-Up on Discussion**
>
> Dear Reviewer,
>
> Thank you for revisiting our submission and increasing your score. If there are any outstanding concerns or points that would benefit from further clarification during the discussion phase, we would be happy to address them.
>
> Sincerely,
>
> Authors of paper 20846.

---

### Official Review · Reviewer_1bsn · 2025-11-05

**Soundness:** 3
**Presentation:** 3
**Contribution:** 2
**Rating:** 4
**Confidence:** 4

**Summary:**

LLM training is memory-bound—mostly by optimizer states—so the paper studies gradient subspaces and finds that (i) most energy concentrates in a small subspace but a meaningful residual remains, (ii) the core subspace’s influence diminishes over time and in deeper layers, and (iii) curvature is near-flat, suggesting geometry-aware methods. Motivated by these observations, it introduces two randomized low-rank algorithms, GrassWalk and GrassJump, which exploit subspace structure to achieve state-of-the-art memory savings and improved pretraining performance on LLaMA-1B/7B.

**Strengths:**

- Identified that most energy concentrates in a small subspace but a meaningful residual remains

- Identified that the core subspace’s influence diminishes over time and in deeper layers

- Identified that curvature is near-flat, suggesting geometry-aware methods

- Proposed two randomized effective low-rank algorithms, GrassWalk and GrassJump

**Weaknesses:**

1. **Subspace observation**. A major contribution of this work is a comprehensive analysis of gradient subspaces during LLaMA pretraining. However, the central result—that the dominance of low-rank subspaces diminishes over time—appears to be implied by the cited work [He et al., 2025], which argues that a random-noise subspace eventually dominates as the effective gradient shrinks. Moreover, [He et al., 2025] recommends using random projection matrices in later stages, consistent with the claim in lines 221–222. The authors should clearly delineate what is new relative to [He et al., 2025].

2. **Insight behind GrassWalk and GrassJump**. I may have missed it, but the paper does not clearly explain why GrassWalk and GrassJump should work. In Section 3, it says that "these observations suggest that the gradient subspace evolves within an almost flat curvature, while there is a considerable portion of energy carried by the bulk component of the gradient." The authors then conclude that "random steps can be advantageous". However, there are many ways to construct the random projection, such as the random Gaussian. I still do not understand why GrassWalk and GrassJump are preferable to a standard Gaussian random projection based on your subspace analysis.

3. **Algorithm design**. It appers to me that the algorithm design may not be sufficiently novel. The paragraph between Line 250-268 seems to be similar to Subtrack++, and the pargraph between 269-307 seems to be similar to Fira. It is better to highlight the novlety more clearly.

4. **Necessity to save optimizer states**. ZeRO-style [R1] sharding already reduces optimizer-state memory by ~1/N per data-parallel replica without changing the Adam mathematical update (and hence maintain the same convergence quality as full-rank Adam). However, it is typically believed that the compressed-state optimizer cannot match the performance of full-rank optimizer. Therefore, I personally think compressing optimizer states is unnecessary due to the existence of the ZeRO technique. I would like to hear the opinion from the authors.

[R1] ZeRO: Memory Optimizations Toward Training Trillion Parameter Models.

**Questions:**

1. Please claify your subspace analysis from that in [He et al., 2025].

2. Please explain the core insight behind GrassWalk and GrassJump—how they are derived from your empirical subspace observations and why they should outperform a Gaussian random projection.

3. Please specify the novel elements of your algorithmic design relative to prior low-rank and subspace-tracking methods.

4. What rank(s) are used in the experiments, and how do your methods compare with full-rank Adam/AdamW in both accuracy and efficiency?

---

> ### Author Response · Authors · 2025-11-22
>
> We thank the reviewer for their thorough evaluation and detailed review. We have addressed the raised concerns and questions below, and have applied the reviewer’s suggestions into our rebuttal revision to strengthen the paper.
>
> > **W1 and Q1. Comparison with GoLore [1]**
>
> Although both works utilize random projections, they arise from fundamentally different perspectives, leading to completely different frameworks.
>
> - GoLore [1] attributes random projections to mitigating an **SVD noise trap** in **later training**. We demonstrate that the critical drop in low-rank energy fraction occurs in the **early training stages** (Figure 1), prior to gradient vanishing. This reveals that crucial information resides in the orthogonal space, which random projections effectively explore.
> - Figure 2 demonstrates the flatness of the **gradient subspace landscape**, distinct from the original gradients. The small singular values indicate that the gradient projection error is minimally sensitive to the direction of subspace updates. This fundamentally differentiates our work from GoLore [1], which addresses diminishing gradients and noise dominance.
>
> > **W2 and Q2. Insight Behind GrassWalk and GrassJump, and Comparison to Gaussian Random Projections**
>
> Our analysis of gradient and gradient subspace geometry confirms: **1)** a significant portion of information lies outside the core subspace (Figure 1), and **2)** an almost flat landscape for gradient subspaces (Figures 2 and 9).
>
> These findings motivate the use of random projections for their two key benefits:
>
> **1) Exploration:** The significant energy in the orthogonal space implies crucial information is often overlooked. Random projections act as a forcing exploration beyond dominant features.
>
> **2) Generalization:** In a flat curvature, random projections facilitate escaping local minima traps, thereby improving the generalization.
>
> Inspired by these benefits, GrassJump utilizes sudden subspace changes for aggressive exploration, while GrassWalk employs controlled, rank-1 random steps to escape local minima more smoothly.
> These distinct properties offer different advantages; for example, GrassWalk shows greater robustness to frequent updates (Table 4), whereas GrassJump performs better with lower ranks (Table 3).
>
> **Comparison to random Gaussian projections:** In GrassJump, we initially generate a random Gaussian matrix and derive a rank-$r$ orthonormal basis via QR decomposition. This is standard practice for orthogonal random projection [8-11], ensuring inherent stability as orthonormal projections preserve norms, angles, and distributions [8-10]. Notably, random Gaussian matrices approximate orthogonality at large scales [2]. We also demonstrated that performing GrassJump without QR orthogonalization yields on-par results in pre-training a 1B model (Table 1; rebuttal revision), reducing wall-time by **~3-4%**, confirming that the random exploration mechanism within our framework is the core contribution, with QR primarily serving theoretical stability.
>
> > **W3 and Q3. Algorithm Design and Differentiation from Prior Works**
>
> Our work introduces a paradigm shift in low-rank optimization by challenging **the core assumption of subspace tracking and adjustment**, employed in prior works, from GaLore [4], to Fira [5], SubTrack++ [3], LDAdam [6], and others [12]. While we share the mathematical foundation (Grassmannian manifolds) of SubTrack++ [3], our algorithmic intent is the inverse to all these prior works.
> - **Exploration over Tracking:** GrassWalk/Jump fundamentally diverge from prior works [1, 3-6, 12], including SubTrack++ [3] by proposing Subspace Exploration over Subspace Adjustment. Derived by our analysis (Figure 1, 2, and 9) we demonstrate that random projection is not merely a computational shortcut, but the natural solution dictated by the underlying optimization geometry, consistently achieving on-par or superior results to SOTA baselines (Tables 1 and 2).
> - **Differentiation from Fira [5]:** Building on literature, we investigate role of information recovery, explored in prior works like Fira [5], APOLLO [2], and SubTrack++ [3], in combination with various subspace selection methods (Figure 3), empirically validating that information recovery along our randomized subspace changes is the superior algorithm design.

---

> > ### Author Response · Authors · 2025-11-22
> >
> > > **W4. Comparison with ZeRO-Style Algorithms**
> >
> > We respectfully disagree that ZeRO renders compressed-state optimizers unnecessary; rather, the two approaches address fundamentally different constraints and are orthogonal directions.
> > - ZeRO is a system-level optimization where memory scales linearly with the number of devices.
> > - Compressed optimizer approaches [1-6, 12] lower the memory footprint on a single hardware.
> >
> > While ZeRO scales out on clusters, low-rank methods [1-6, 12] scaling up model sizes on constrained hardware. In fact, they can help improve the communication overhead of distributed methods.
> >
> > **Performance of low-rank methods:** In fact, recent low-rank methods, including GrassWalk and GrassJump (Table 1; rebuttal revision), SubTrack++ [3], Fira [5], LDAdam [6], and FRUGAL [7], have shown on-par or better performance compared to full-rank training, as we will describe in next section.
> >
> > > **Q4. Rank of Subspace and Comparison with Full-Rank Training**
> >
> > For Llama-1B, a rank of 512, and for Llama-7B, a rank of 1024 was used. The hyperparameters are reported in Table 7 of the rebuttal revision.
> >
> > We demonstrated on-par or surpassing performance compared to full-rank training (Table 1; rebuttal revision). This phenomenon, also reported in prior works like SubTrack++ [3], LDAdam [6], and FRUGAL [7],  is primarily attributed to the regularization effect of low-rank approaches [6, 3].
> >
> > In addition, GrassWalk and GrassJump maintain GaLore’s [4] memory (reducing full-rank memory bottleneck) with comparable computational efficiency with fastest baselines such as APOLLO [2], and SubTrack++ [3], introducing neglectable computational overhead.
> >
> > *Table 1. Comparing Full-Rank Training with GrassWalk and GrassJump*
> > |Method|Evaluation Loss |Memory (GB)|Wall-Time (m)|
> > |:-|:-|:-|:-|
> > |FullRank|4.10|35.2|417.0|
> > |GrassWalk|3.86|32.0|418.6|
> > |GrassJump |3.87|32.1|432.5|
> >
> > ---
> >
> > We thank the reviewer for their thoughtful feedback. We have incorporated all requested experiments and clarifications into the revised rebuttal version. We hope that these additions address the reviewer’s concerns, and we would be grateful if the reviewer could consider updating their evaluation accordingly.
> >
> > ---
> > [1] Chen et al., 2025. *Greedy Low-Rank Gradient Compression for Distributed Learning with Convergence Guarantees.*
> >
> > [2] Zhu et al., 2025. *APOLLO: SGD-like Memory, AdamW-level Performance.*
> >
> > [3] Rajabi et al., 2025. *SubTrack++: Gradient Subspace Tracking for Scalable LLM Training.*
> >
> > [4] Zhao et al., 2024. *GaLore: Memory-Efficient LLM Training by Gradient Low-Rank Projection.*
> >
> > [5] Chen et al., 2024. *Fira: Can We Achieve Full-rank Training of LLMs Under Low-rank Constraint?*
> >
> > [6] Robert et al., 2025. *LDAdam: Adaptive Optimization from Low-Dimensional Gradient Statistics.*
> >
> > [7] Zmushko et al., 2024. *FRUGAL: Memory-Efficient Optimization by Reducing State Overhead for Scalable Training.*
> >
> > [8] Andrecut, 2024. *Residual Random Neural Networks.*
> >
> > [9] Banf and Hartwig, 2021. *Reasonable Effectiveness of Randomness in Scalable and Integrative Gene Regulatory Network Inference and Beyond.*
> >
> > [10] Murray et al., 2023. *Randomized Numerical Linear Algebra : A Perspective on the Field With an Eye to Software.*
> >
> > [11] Zhao et al., 2015. *Efficient Clustering on Riemannian Manifolds: A Kernelised Random Projection Approach.*
> >
> > [12] Liang et al., 2024. *Memory-Efficient LLM Training with Online Subspace Descent.*

---

> ### Author Response · Authors · 2025-11-28
> **Follow-Up on Discussion**
>
> Dear Reviewer,
>
> We wanted to once again appreciate your detailed feedback. Also, if any of our earlier clarifications require further elaboration, or if additional concerns remain, we would be glad to provide more information while the discussion window is still open.
>
> Sincerely,
>
> Authors of paper 20846.

---

### Official Review · Reviewer_11mw · 2025-11-07

**Soundness:** 2
**Presentation:** 3
**Contribution:** 2
**Rating:** 4
**Confidence:** 3

**Summary:**

The paper studies low-rank gradient projection for memory-efficient LLM training and proposes two randomized subspace methods, GrassWalk and GrassJump. The authors  analyze gradient subspace dynamics in pretraining, observing that (i) a core low-rank subspace captures a substantial fraction of gradient energy but its share decreases over time and in deeper layers, and (ii) the gradient subspace evolves under nearly flat curvature. These observations motivate randomized subspace exploration combined with (a) projection-aware Adam state updates when the basis changes and (b) recovery scaling to re-inject information lost by low-rank projection into the full-rank update.

**Strengths:**

1. Clear conceptual framing  and insightful analysis of gradient subspace dynamics (diminishing core energy; near-flat curvature) that motivates randomized exploration.


2. Proposed GrassWalk and GrassJump avoid expensive per-step SVD and include principled AO and RS to mitigate basis changes and projection loss.

3. Competitive short-budget results on LLaMA-1B  (10k steps) and 7B, beating strong baselines at similar memory/time.

**Weaknesses:**

1. The experiments are short-run, limited to 10k training steps for LLaMA-1B and 7B. While results show modest early-phase gains, such improvements at this scale do not guarantee similar behavior in longer or full pretraining regimes, where optimization dynamics and curvature can change substantially. Longer runs or additional model families are needed to confirm scalability and stability.

2. The evaluation is restricted to training/eval loss, without downstream or zero-shot benchmarks to test whether early-loss advantages translate to useful model quality.

3. Sensitivity to key hyperparameters such as rank r, update interval T, and AO/RS controls is not explored.

4. Evidence is confined to LLaMA models. Validating on diverse architectures would better demonstrate generality.

**Questions:**

1. Please report total tokens processed for each run. Also provide sequence length, effective global batch size, and grad-accum steps.


2. Can you extend comparisons to ≥50k–100k steps (or a few billion tokens) with the same baselines to test if the ranking at 10k steps persists?

3. Provide ablations over rank r, update interval T, AO details, and RS limiter ζ . How sensitive are the results to these choices?

4. The analysis argues for near-flat curvature and diminishing core energy. Can you provide quantitative metrics alongside training curves to show correlation with method performance?

---

> ### Author Response · Authors · 2025-11-22
>
> We thank the reviewer for their thorough evaluation and for finding our analysis insightful. We have addressed the raised concerns and questions below, and have applied the reviewer’s suggestions into our rebuttal revision to strengthen the paper.
>
> > **W1, Q2, and W4. Longer Runs and Additional Model Families**
>
> Per reviewer suggestion, we compared our methods, GrassWalk and GrassJump, with our best performing baseline, SubTrack++ [1], over longer runs and other model families:
>
> - Extended the pre-training of **Llama-7B** to **100k iterations** (Table2, Figure 8; rebuttal revision)
> - Pre-trained **Qwen2-1.5B** architecture for 10k iterations (Table 1; rebuttal revision)
>
> The results demonstrate the ranking persists, confirming the stability and scalability of our methods across different architectures and training budgets.
>
> *Table 1. Pre-training Llama-7B for 100k iterations.*
> | Method|Evaluation Loss|
> | :-| :-|
> |SubTrack++ |3.37|
> |GrassWalk| 3.37|
> |GrassJump| 3.34|
>
> *Table 2. Pre-training Qwen2-1.5B for 10k iterations.*
> |Method|Evaluation Loss|
> | :-| :-|
> |AdamW|4.84|
> |SubTrack++|4.70|
> |GrassWalk|4.68|
> |GrassJump|4.66|
>
> > **Q1. Processed Tokens and Training Setup**
>
> Hyperparameters are now reported in **Appendix D** of the revision, and we are reporting them here for your reference:
>
> - **Llama-1B:** 10k iterations with batch-size 8, gradient accumulation factor 2, and sequence length 256 leads to ~41M tokens during training.
> - **Llama-7B:** 100k iterations with batch-size 4, gradient accumulation factor 4, and sequence length 256 leads to ~410M tokens during training.
>
> > **W2. Zero-Shot Benchmarks**
>
> We follow the standard evaluation protocol in low-rank training work, reporting pre-training validation loss or perplexity, as done in GaLore [3], Fira [2], LDAdam [4], SubTrack++ [1], FRUGAL [5], and others [6,7]. Zero-shot benchmarks like MMLU-Pro require substantially longer pre-training and fall outside the settings and budget used in these studies. Using the established protocol ensures fair and directly comparable results.
>
> > **W3 and Q3. Hyperparameter Ablations**
>
> Per reviewer suggestion, we report the results of ablations regarding key hyperparameter settings. Also:
>
> - The AO component introduces no additional hyperparameters.
> - The impact of integrating AO and RS components with various subspace adjustment methods is systematically detailed in the ablation studies in **Section 5** and **Figure 3**.
> - Unless noted otherwise, the experiments employed a Llama-1B architecture pre-trained for 10k iterations, utilizing a subspace rank of 512 and a subspace update interval of 200.
>
> ### **Subspace Rank (Table 3; rebuttal revision)**
> The following table examines the sensitivity of GrassWalk and GrassJump to subspace rank with subspace update interval 200. While a higher rank yields better loss, the marginal benefit is limited. Notably, in GrassJump, the reduced dependency on a specific subspace allows for effective performance even with significantly lower ranks.
>
> *Table 3. Subspace rank ablation*
> |Method|256|512|1024|
> |:-|:-|:-|:-|
> |GrassWalk|4.62|4.52|4.43|
> |GrassJump|4.54|4.50|4.45|
>
> ### **Update interval T (Table 4; rebuttal revision):**
> While increased update frequency generally improves final performance, overly frequent GrassJump updates degrade results. This suggests that drastic subspace changes disrupt optimizer convergence, requiring sufficient iteration time.
>
> *Table 4. Subspace update interval ablation*
> |Method|20|50|100|200|500|1000|2000|
> |:-|:-|:-|:-|:-|:-|:-|:-|
> |GrassWalk|3.72|3.79|3.86|3.98|4.25|4.46|4.68|
> |GrassJump|4.33|4.34|3.87|3.96|4.15|4.75|4.89|
>
> ### **Norm-growth limiter ζ (Table 5, Figure 7; rebuttal revision):**
> As shown in Figure 7, this parameter acts similarly to a gradient-clipping coefficient [2], regulating training spikes. Results demonstrate that mitigating these spikes, while avoiding excessive suppression of the recovered signal (e.g., (ζ = 0.1)) has a substantial impact on the final evaluation loss.
>
> *Table 5. Norm-Growth Limiter Ablation*
> |Method|1.01|0.1|10|off|
> |:-|:-|:-|:-|:-|
> |GrassWalk|3.29|3.44|3.41|3.42|
> |GrassJump|3.30|3.45|3.44|3.44|

---

> > ### Author Response · Authors · 2025-11-22
> >
> > > **Q4. Quantitative Metric Diminishing Core Energy and Curvature**
> >
> > Figures 1 and 2 are demonstrating the gradient space properties, and describing the underlying subspace optimization dynamics, which are inherent and independent of the training method, which can be GaLore [3], SubTrack++ [1], APOLLO [8], or any other methods [1-8]. Therefore, they do not have a direct correlation with the method performance. Nevertheless, these observations can be quantified independently:
> >
> > - In Figure 1, we report the fraction of gradient energy captured by the low-rank representation, as $\frac{|| \widetilde{G}_t ||_F}{||G_t||_F }$.
> > - In Figure 2, we can summarize the flatness into a single scalar metric by computing the Frobenius norm of these singular values (Figure 9, rebuttal revision). As illustrated, almost all layers exhibit extremely small values, indicating that the subspace estimation error has very low sensitivity in every direction.
> >
> > These properties are inherent, challenging the dominant core assumption of prior works [1-4, 6, 7], and explaining why not only randomized methods can work, but also embedded in our framework, outperform all SOTA baselines (Tables 1 and 2). We have clearly labeled these metrics in the plots and the accompanying text in the rebuttal revision.
> >
> > ---
> >
> > We thank the reviewer for their thoughtful feedback. We have incorporated all requested experiments and clarifications into the revised rebuttal version. We hope that these additions address the reviewer’s concerns, and we would be grateful if the reviewer could consider updating their evaluation accordingly.
> >
> > ---
> >
> >
> > [1] Rajabi et al., 2025. *SubTrack++: Gradient Subspace Tracking for Scalable LLM Training.*
> >
> > [2] Chen et al., 2024. *Fira: Can We Achieve Full-rank Training of LLMs Under Low-rank Constraint?*
> >
> > [3] Zhao et al., 2024. *GaLore: Memory-Efficient LLM Training by Gradient Low-Rank Projection.*
> >
> > [4] Robert et al., 2025. *LDAdam: Adaptive Optimization from Low-Dimensional Gradient Statistics.*
> >
> > [5] Zmushko et al., 2024. *FRUGAL: Memory-Efficient Optimization by Reducing State Overhead for Scalable Training.*
> >
> > [6] Liang et al., 2024. *Memory-Efficient LLM Training with Online Subspace Descent.*
> >
> > [7] He et al., 2025. *Subspace Optimization for Large Language Models with Convergence Guarantees.*
> >
> > [8] Zhu et al., 2025. *APOLLO: SGD-like Memory, AdamW-level Performance.*

---

### Author Response · Authors · 2025-12-03
**Summary After Rebuttal Interruption**

**Dear ICLR AC Chairs,**

Due to the OpenReview data-leak issue and the rollback of reviewer scores, we are providing a concise summary of the issues raised and how we fully addressed them in the rebuttal.

**1. Differentiation from prior work** (Reviewers 1bsn, X2EV, VgBp)

We clarified that our work challenges the foundational assumption behind gradient low-rank methods that fixated on the need to adjust the underlying subspace, employed in many recent prior works [1-5]. Our analysis of **gradient-energy distribution** and its **subspace curvature** shows **why** randomized updates are effective [6, 7] and **when** they can surpass sophisticated tracking methods (Figures 1 to 3). As shown in Tables 1 and 2, GrassWalk and GrassJump achieves the best performance in all settings surpassing even the strongest, most disciplined baselines [3, 4] while remaining highly computationally efficient.

**2. Additional analyses and experiments**

 In response to reviewer requests, we conducted several additional analyses: extended ablations to assess sensitivity to various hyperparameters (Reviewers 11mw, VgBp); confirmation of our gradient-behavior observations across different subspace ranks (Reviewer X2EV); evaluation of the generalization of our methods across broader model families and longer training regimes (Reviewer 11mx); and multi-seed experiments (Reviewer X2EV). All reviewer questions were addressed thoroughly, and the rebuttal has been updated accordingly.

----

Reviewer X2EV increased their score after reading our rebuttal. Although the remaining reviewers were unable to revisit our updated responses due to the abrupt interruption, their initial scores were positive, and we are confident they would find the revisions fully satisfactory.

Sincerely,

Authors of paper 20846.

-----

[1] Zhao et al., 2024. GaLore: Memory-Efficient LLM Training by Gradient Low-Rank Projection.

[2] Chen et al., 2024. Fira: Can We Achieve Full-rank Training of LLMs Under Low-rank Constraint?

[3] Rajabi et al., 2025. SubTrack++: Gradient Subspace Tracking for Scalable LLM Training.

[4] Robert et al., 2025. LDAdam: Adaptive Optimization from Low-Dimensional Gradient Statistics.

[5] Liang et al., 2024. Memory-Efficient LLM Training with Online Subspace Descent.

[6] Chen et al., 2025. Greedy Low-Rank Gradient Compression for Distributed Learning with Convergence Guarantees.

[7] Zhu et al., 2025. APOLLO: SGD-like Memory, AdamW-level Performance.

---

### Meta-Review · Area_Chair_26t6 · 2025-12-20

**Summary:**

This paper studies low-rank gradient training for memory-efficient LLM pretraining, and proposes two randomized subspace update strategies based on Subtrack++. All reviewers' initial comments are consistently negative. The main concern is regarding the similarity with prior works such as Subtrack++ and randomized subspace methods for memory efficient pretraining. Particularly, the paper is written in a very similar style as Subtrack++ and it is worth compressing the contents that are not part of the contributions. In addition, the benefit over the baselines (as reported in the experiments) seems marginal.

**Reviewer Concerns:**

Concerns regarding additional experiments, ablations and sensitivity analysis have been addressed in the rebuttal. The concerns regarding novelty and positioning of the paper among prior works are still valid.

**Reviewer Scores:**

Reviewer 11mw, 1bsn, VgBp that gave 4 may still remain borderline. Reviewer X2EV that gave 2 may remain negative.

---

### Decision · Program_Chairs · 2026-01-26

Reject